# The Multifaceted Output of c-Jun Biological Activity: Focus at the Junction of CD8 T Cell Activation and Exhaustion

**DOI:** 10.3390/cells9112470

**Published:** 2020-11-13

**Authors:** Athanasios G. Papavassiliou, Anna Maria Musti

**Affiliations:** 1Department of Biological Chemistry, Medical School, National and Kapodistrian University of Athens, 11527 Athens, Greece; papavas@med.uoa.gr; 2Department of Pharmacy, Health and Nutritional Sciences, University of Calabria, 87036 Rende, Italy

**Keywords:** c-Jun, AP-1, CD8 T cell, differentiation, exhaustion, transcriptional, epigenetic, immunotherapy

## Abstract

c-Jun is a major component of the dimeric transcription factor activator protein-1 (AP-1), a paradigm for transcriptional response to extracellular signaling, whose components are basic-Leucine Zipper (bZIP) transcription factors of the Jun, Fos, activating transcription factor (ATF), ATF-like (BATF) and Jun dimerization protein 2 (JDP2) gene families. Extracellular signals regulate c-Jun/AP-1 activity at multiple levels, including transcriptional and posttranscriptional regulation of c-Jun expression and transactivity, in turn, establishing the magnitude and the duration of c-Jun/AP-1 activation. Another important level of c-Jun/AP-1 regulation is due to the capability of Jun family members to bind DNA as a heterodimer with every other member of the AP-1 family, and to interact with other classes of transcription factors, thereby acquiring the potential to integrate diverse extrinsic and intrinsic signals into combinatorial regulation of gene expression. Here, we review how these features of c-Jun/AP-1 regulation underlie the multifaceted output of c-Jun biological activity, eliciting quite distinct cellular responses, such as neoplastic transformation, differentiation and apoptosis, in different cell types. In particular, we focus on the current understanding of the role of c-Jun/AP-1 in the response of CD8 T cells to acute infection and cancer. We highlight the transcriptional and epigenetic regulatory mechanisms through which c-Jun/AP-1 participates in the productive immune response of CD8 T cells, and how its downregulation may contribute to the dysfunctional state of tumor infiltrating CD8 T cells. Additionally, we discuss recent insights pointing at c-Jun as a suitable target for immunotherapy-based combination approaches to reinvigorate anti-tumor immune functions.

## 1. Introduction

Active AP-1 is a dimeric complex whose components are basic-Leucine Zipper (bZIP) transcription factors (TFs) of the Jun, Fos, activating transcription factor (ATF), ATF-like (BATF) and Jun dimerization protein 2 (JDP2) gene families, binding DNA as an obligate dimer through their bZIP domain, with DNA-binding preferences being partner-specific [1]. Except for the Fos subfamily, AP-1 subunits can form all possible dimer combinations, expanding the repertoire of cognate sites that might be targeted by different heterodimers [1,2]. Furthermore, AP-1 dimers can interact with other classes of inducible TFs, with which they form ternary complexes that bind to composite DNA motifs [3]. In particular, ternary complexes formed by Jun/Fos and Jun/BATF heterodimers with the nuclear factor of activated T cells (NFAT) and interferon responsive factor (IRF) families of TFs, respectively, have been shown to control regulatory networks in CD8 T cells [4,5,6,7]. 

Depending on the strength and extent of antigen (Ag)-challenge, costimulatory and cytokine signals, CD8 T cells can undergo distinct cell fate programs, differentiating in either effector or memory cells [7]. In contrast, during cancer and chronic infection, antigen-primed CD8 T cells undertake a dysfunctional state, generically referred as T cell exhaustion [8]. 

Multiple studies have shown that NFAT and canonical AP-1, mainly c-Jun/c-Fos heterodimer, function downstream of T-cell receptor (TCR) and CD8 costimulatory receptor, and synergistically induce the expression of IL-2 and other effector molecules in Ag-primed CD8 cells [4,9,10,11,12,13,14,15,16,17,18]. Additionally, Jun/BATF/IRF4 ternary complexes promote chromatin accessibility at the enhancers of genes coding for lineage-specific transcription factors and effector genes in effector CD8 T cells [6,7,19,20]. 

On the other hand, studies on transcriptional landscapes of exhausted CD8 T cells in chronic infection and cancer have shown that the early transcriptional pathway driving exhaustion is initiated by NFAT activation in absence of c-Jun/AP-1 cooperation [21,22,23]. This transcriptional program is characterized by the gradual downregulation of NFAT/AP-1-target genes and the induction of immune checkpoint inhibitor receptors (IRs), such as programmed cell death protein 1 (PD-1), cytotoxic T-Lymphocyte Antigen 4 (CTLA-4) and T-cell Immunoglobulin domain and mucin domain 3 (TIM-3) [21,22,24]. Further studies have shown that partnerless NFAT activates secondary TFs, such as the HMG-box transcription factor TOX and nuclear receptor 4a (NR4A), in turn promoting chromatin accessibility at genes driving CD8 T cell exhaustion in chronic infection and cancer [25,26,27,28,29]. These same TFs were shown to drive exhaustion also in CD8 T cells engineered to express chimeric antigen receptors (CAR T cells) for the targeting of solid tumors [29,30,31]. 

The elucidation of the core transcriptional pathway initiating T cell exhaustion in different settings has opened the road to more effective immunotherapeutic strategies, aiming to reinvigorate T cell responses. In this regard, recent studies have shown that CAR T cells deficient in either TOX or NR4A exhibit a gene expression profile characteristic of CD8 effector T cells, with open chromatin regions enriched for both NF-κB and c-Jun/AP-1 binding motifs [29,31]. Conversely, Lynn at al. have shown that CAR T cells engineered to overexpress c-Jun are resistant to exhaustion, suggesting that the JNK/c-Jun pathway is a suitable target for immunotherapy-based combination approaches to reinvigorate anti-tumor immune functions [30]. 

Following a description of the principal and most current insights into the regulation and function of the c-Jun/AP-1 transcription factor in different types of mammal cells, this review focus on the current understanding of c-Jun/AP-1 function in the immune response of CD8 T cell to acute infection and cancer. In particular, we discuss the epigenetic and transcriptional mechanisms through which c-Jun/Ap1 contributes to the productive immune response of CD8 T cells, and how its deregulation in tumor reactive CD8 T cells may facilitate T cell dysfunction. 

## 2. Function and Regulation of the c-Jun/AP-1 Transcription Factor

### 2.1. AP-1 Complexity 

Activator protein-1 (AP-1) is a family of ubiquitous dimeric transcriptional complexes. AP-1 activity is involved in a variety of cellular processes, and has been also implicated in molecular mechanisms underpinning several diseases [1]. The role of these well-described transcriptional complexes is to integrate signals from diverse extracellular cues and coordinate the respective cellular response [1]. This role applies to various physiological processes, but has been also documented in a broad spectrum of pathophysiologies, including maladies such as asthma, psoriasis, rheumatoid arthritis and predominantly cancer [1]. Therefore, AP-1 protein components represent attractive therapeutic targets [32].

AP-1 collectively describes a group of structurally and functionally related proteins that share an evolutionary conserved basic DNA-binding domain combined with a leucine zipper region (bZIP domain). The leucine zipper mediates dimerization of the AP-1 protein components, which is necessary for the basic domain to bind to cognate DNA motifs. AP-1 transcriptional complexes mainly consist of members of the Jun (c-Jun, JunB, JunD) and the Fos (c-Fos, FosB, Fra-1, Fra-2) protein families. Members of the activating transcription factors (ATF) (ATF-2, ATF-3/LRF1, ATF-4, ATF-5, ATF-6B, ATF-7, BATF, BATF-2, BATF-3), JDP (JDP-1 and JDP-2) and MAF (c-MAF, MAFA, -B, -F, -G, -K, and Nrl) protein subfamilies may also participate in complexes with AP-1 proteins and share the diverse DNA target sequences. However, Jun and Fos proteins are the principal representatives and most studied AP-1 proteins. AP-1 regulates both basal and inducible transcription of various genes harboring AP-1 sites (consensus sequence 5′-TGAG/CTCA-3′), which are known as TPA-response elements (TREs). Certain AP-1 complexes also show affinity for cAMP-response elements (CREs) encompassing the 5′-TCACGTCA-3′ sequence. Recent genome-wide developmental and cancer studies reveal that AP-1 probably exerts its actions through binding to distal transcriptional enhancers, rather than to promoter regions [1,33,34,35,36,37]. It is noteworthy that, while Jun proteins have the capacity to either homo- or heterodimerize, Fos proteins can only form heterodimers with Jun proteins, thereby assisting their DNA-binding potential [1,2].

### 2.2. AP-1 Functions: From Tissue Differentiation to Cancer Progression and Immune Response

Regarding the cellular functions where AP-1 plays a key role, it is well documented that AP-1 transcriptional complexes are engaged in processes of proliferation, differentiation, inflammation and apoptosis [38]. Recent data highlight the role of AP-1 in development and differentiation, where experiments in primary mouse embryonic fibroblasts (MEFs) render AP-1 a central factor entailed in the process of enhancer selection, a prerequisite for MEFs to follow differentiating programs. AP-1 co-operates with cell-type specific transcription factors for enhancer selection and recruits the SWI/SNF (BAF) remodeling complex to form accessible chromatin structure [36]. Differentiation of the developing cells of the skin epidermis requires the specific regulatory enhancer 923 on the epidermal differentiation complex locus, which binds c-Jun/AP-1 in order to promote the process [35]. Corroborating evidence reveals that AP-1 also controls bone formation and bone homeostasis through several mechanisms, among which is the positive regulation of the Fra-2/AP-1 complex [39], while it also controls blood tissue development through regulation of the balance between vascular smooth muscle and hematopoietic cells [40]. AP-1 and c-Fos, in particular, are also implicated in neuronal function through modification of chromatin assembly [41,42]. Other differentiation-associated functions include the AP-1 binding to enhancers that modulate macrophage expression and CD8 T cell differentiation [40,43].

AP-1 transcription factors are master regulators of oncogenic events and influence the expression of a series of regulators of cell proliferation, migration and survival, which are critically involved in cancer development and metastasis. Whilst c-Jun oncoprotein seems to drive proliferation and migration in human cancers, at the same time, there are Jun (JunB and JunD), Fos and ATF proteins that have been suggested to exert tumor suppressor function. Whether AP-1 functions as a tumor promoter or a tumor suppressor depends on the antagonistic activity of diverse AP-1 proteins, on the type of the malignancy, the stage of the disease and, finally, the genetic landscape of the tumor [44,45]. The contribution of AP-1 activity to the hallmarks of cancer is well established by numerous studies, however recent experimental data highlight new aspects of AP-1 activity. With regard to proliferation, cancer cells not only require high levels of AP-1 components such as c-Jun and Fra-1 throughout the cell cycle, but they also maintain an autocrine/paracrine loop of high Fra-2/AP-1 expression, in order to enhance proliferation and migration for themselves and their microenvironment [46]. There is also further elucidation of the mechanisms via which AP-1 drives oncogenic growth and metastasis. A number of cancer studies suggest that this is achieved through positive selection of enhancer elements by AP-1, either on its own or in synergy with other transcription factors [33,34,37,47,48]. 

AP-1 holds a crucial role in several aspects of the immune system, such as T-cell activation, T-helper (Th) differentiation, T-cell anergy and exhaustion [7,14,21]. It has been also found that AP-1 induces transcriptionally the expression of co-inhibitory immune checkpoints (PD-1, PD-L1) and the expression of the *FOXP3* gene locus, which is a key modulator of regulatory T cells (Treg); therefore, AP-1 is implicated in the efficacy of anti-tumor T-cell responses and immunotherapy [49]. In this vein, AP-1 recently emerged to regulate mechanisms of drug resistance to formerly successful treatments. Most resistant mechanisms to drugs that target mutated molecules are generated from secondary mutations; however, there are cases with no genetic cause presenting non-genetic rare cell variability. C-Jun and/or AP-1 mediate signaling pathways that ultimately lead to epigenetic reprogramming in these cells and confer permanent drug resistance [50]. 

### 2.3. Levels of c-Jun/AP-1 Regulation

The activity of c-Jun/AP-1 proteins can be regulated in a multi-level manner. It depends on the abundance of AP-1 proteins and composition of the complex itself, modulation of transcription of genes that encode AP-1 subunits, mRNA turnover and protein stability, post-translational modifications and interactions with other transcription factors and co-factors [44].

#### 2.3.1. Dimer Composition

At first, the composition of the dimers themselves differentiates the transcriptional capacity of the complex. Thus, Jun/Fos dimers exhibit higher DNA-binding affinity than Jun/Jun dimers, as well as more active stimulated transcriptional ability [1,51]. Furthermore, Jun and Fos have different transactivation potentials. c-Jun, c-Fos and FosB proteins harbor an N-terminal transactivation domain, whereas JunB, JunD, Fra-1, Fra-2 and FosB2 demonstrate low transactivation activity [52,53]. Although Jun homo- and heterodimers are ubiquitously expressed, each component presents unique cell- and tissue-specific distribution and trans-targeted stability, facts that further perplexes the mode of their activity [53,54,55].

#### 2.3.2. Transcriptional and Post-Translational Modifications

The transcriptional activity of c-Jun/AP-1 is modulated by a wide variety of cellular and extracellular cues, including growth factors, bacterial and viral infection, cytokines, chemokines, hormones, ultraviolet (UV) irradiation, cellular and environmental stresses (e.g., hypoxia), thereby affecting the homeostasis of AP-1 within cells (Figure 1A). These cellular and environmental stimuli can result to altered c-Jun/AP-1 capacity of forming dimers, binding to DNA and activating gene transcription (Figure 1A) [38,54,56]. In particular, most cells contain endogenous, basal levels of c-Jun expression. c-Jun abundance is further enhanced by induction of the *c-jun* promoter through the TRE element, which favors binding of c-Jun/ATF-2 heterodimers. Therefore c-Jun/AP-1 has the capacity of autoregulation, forming positive and negative feedback loops (Figure 1A) [38,57,58,59].

c-Jun mRNA status is regulated by CAP-dependent and CAP-independent translational mechanisms via the internal ribosome entry segments (IRES) [60]. c-Jun/AP-1 abundance is also regulated through epigenetic changes of the mRNA status. The long non-coding RNA (lncRNA) HOTAIRM1 upregulates the JNK/c-Jun pathway, therefore promoting osteogenic differentiation in mesenchymal stem cells via a c-Jun-mediated regulation of the bone-specific transcription factor runt-related transcription factor 2 (Runx2) [61]. 

c-Jun expression is also regulated through micro-RNAs (miRNAs). MiR-10b has been found upregulated in metastatic breast cancer cells in conjunction with c-Jun elevation. This is a MAPK-independent event that occurs through RhoC and NF1 of the Rho-associated protein kinase (ROCK) pathway, which controls cytoskeletal dynamics [62].

Post-translational modifications are decisive for c-Jun/AP-1 regulation and mainly refer to phosphorylation by kinases that belong to the mitogen-activated protein kinases (MAPKs) family. The family comprises extracellular signal-regulated kinase (ERK), c-Jun N-terminal kinase (JNK), p38, and c-Fos-regulating kinase (FRK) [38]. MAPKs are serine/threonine kinases that become activated by the aforementioned cellular and extracellular cues and control the concentration, as well as the transactivating efficiency of Jun, Fos and ATF proteins [38]. C-Jun in particular is the most prominent transcriptional activator among the AP-1 proteins. Its transcriptional activity is enhanced via signal transduction pathways upon stimulation by growth factor and stress cues. This enhanced via signal transduction pathways upon stimulation by growth factor and stress cues. This leads to alterations in the phosphorylation state of c-Jun within minutes after the application of the stimulus, and the event takes place independently of de novo protein synthesis in the already present c-Jun protein [53,54,63]. 

In addition to the phosphorylation/dephosphorylation processes that have been extensively studied, post-translational acetylation and sumoylation of c-Jun negatively control its transactivity (Table 1) [64,65,66,67].

### 2.4. Regulation of c-Jun Activity by N-Terminal and C-Terminal Phosphorylation

Focusing on the regulation of c-Jun via post-translational phosphorylation/dephosphorylation events, it is established that a variety of extracellular stimuli such as genotoxic stresses (e.g., UV irradiation), pro-inflammatory cytokines, growth factors and hormones trigger c-Jun activity mostly through JNK (Figure 1A). Following activation, JNK translocates to the nucleus and phosphorylates c-Jun at Ser63/73 and Thr91/93 residues, which are two positive regulatory clusters residing within its N-terminal transactivation domain [68,69,70,71]. This event leads to the homodimerization of c-Jun or heterodimerization with c-Fos, thereby activating transcription (Figure 1A) [38,45,54,80,81]. JNK also phosphorylates (on Thr69/71 residues) and activates ATF-2, thus leading to ATF-2/c-Jun heterodimerization and indirect c-Jun activation. This subsequently potentiates binding of the dimer to alternative AP-1-binding sites in the *c-jun* promoter, further augmenting the c-Jun transcriptional effect [45,59].

Although the molecular mechanism(s) underlying the ability of JNK to regulate c-Jun activity remain partly elusive, it is clear that this process involves modulation of physical interactions with a network of co-activators and/or co-repressors that regulate histone acetylation and, ultimately, the euchromatin/heterochromatin status [45,52,82]. N-terminally phosphorylated c-Jun by JNK exhibits higher affinity for the transcriptional co-activating complex cAMP-response element binding protein (CREB)-binding protein (CBP)/p300 and enables gene transcription [83,84,85]. JNK-phosphorylated c-Jun also recruits histone deacetylases (HDACs), for example HDAC1, in order to promote neuropathic pain after nerve injury [86]. It also interacts with the transcription factor 4 (TCF-4), forming a complex, along with *β*-catenin and potentiates transcription of *c-jun*, in order to promote intestinal tumorigenesis [87]. The suggested mechanism by which specific N-terminal phosphorylation of c-Jun enhances gene transcription is by inhibiting previously established repression. Repression of gene transcription is achieved because unphosphorylated c-Jun recruits and interacts with methyl-CpG-binding domain protein 3 (Mbd3) and, subsequently, with the nucleosome remodeling deacetylase (NuRD) repressor complex, thereby hampering transcription. This suppressive function is reversed by JNK-mediated phosphorylation of c-Jun [88].

On the other hand, c-Jun harbors three phosphorylation sites (residues 227–252) at the C-terminal DNA-binding domain that, when phosphorylated, reduce the DNA-binding affinity of c-Jun, hence, suppressing its transcriptional activity [54,76,77]. Specifically, c-Jun is primarily phosphorylated at Ser243 and subsequent phosphorylation at Thr239 is catalyzed by glycogen synthase kinase-3 (GSK-3) [70,74]. GSK-3 phosphorylates c-Jun and keeps it in a non-binding state [89]. An alternative MAPK pathway that includes activation of ERK leads to GSK-3 phosphorylation through intermediate phosphorylation/activation of p70S6 kinase (p70S6K), thus, rendering GSK-3 inactive. Inactive GSK-3 is not able to phosphorylate c-Jun at its C-terminus, thereby restoring DNA-binding affinity and c-Jun transcriptional activity [53]. Furthermore, c-Jun phosphorylation at Ser63/73 and Thr91/93 sites results in dephosphorylation of Thr239, unraveling an indirect effect of c-Jun N-terminal phosphorylation on DNA-binding activity [70,71]. 

N-terminal phosphorylation frequently acts in concert with the C-terminal phosphorylation status of c-Jun. Notably, GSK-3 inactivation promotes apoptosis in pancreatic cancer cells, a process that also requires JNK activation [90]. Moreover, GSK-3*β* can negatively regulate growth factor-mediated activation of JNK [75]. 

### 2.5. Impact of c-Jun N-Terminal Phosphorylation on c-Jun Ubiquitine-Depedendent Degradation 

c-Jun was among the first oncogenic transcription factors being shown to be degraded by the ubiquitin/proteasome system [78]. Furthermore, Musti et al. have shown that in NIH3T3 fibroblasts JNK1-dependent phosphorylationof of c-Jun N-terminal domain inhibits c-Jun polyubiquitination, hence increasing protein stability [91]. Later on, Nateri and collaborators discovered that the Fbw7 was the major ubiquitin ligase targeting c-Jun for proteasome-dependent degradation in neurons [79]. In contrast with the previous study in NIH3T3 fibroblasts, Nateri and collaborators also showed that c-Jun ubiquitination by Fbw7 required N-terminal phosphorylation by JNK at Ser63/73 and Thr91/93 sites, rendering these sites part of an N-terminal degron-domain allowing Fbw7 binding [79]. However, a further study demonstrated that depending on which JNK isoform is mainly active in the nucleus, c-Jun N-terminal phosphorylation may either inhibit or induce c-Jun degradation: in fibroblasts and MEF, nuclear c-Jun is mainly phosphorylated by JNK1, which results in a significant increase of c-Jun stability, whilst JNK2 facilitates ubiquitin binding to c-Jun [92]. In neurons, JNK1 is mostly cytosolic, whereas JNK2/JNK3 are the main JNK isoforms traslocating to the nucleus [93], suggesting that JNK2 might be the main JNK isoform targeting nuclear c-Jun for Fbw7-dependent polyubiquitinatiIthe brain. However, more studies are required to elucidate which JNK isoform targets c-Jun for ubiquitin-dependent degradation in neurons. 

On the other hand, Fbw7 binding to its target proteins (e.g., c-Myc and cyclin E) requires phosphorylation of a GSK-3 specific site within the degron-domain [94]. Such a dragon domain is present within the C-terminus of c-Jun, containing the GSK-3 specific site Thr239, whose phosphorylation was shown to be necessary for the FBw7-dependent ubiquitination of c-Jun in several cell lines [74]. Notably, this C-terminal degron was shown to regulate c-Jun stability only in the absence of mitogenic signaling [74]. This finding suggests that the effect of JNK1-dependent c-Jun N-terminal phosphorylation on c-Jun ubiquitination (observed by Musti et al.) might be secondary to dephosphorylation of Thr293, which is triggered by c-Jun N-terminal multisite phosphorylation [70,71].

Interestingly, the opposite effect of JNK1 and JNK2 on c-Jun stability mirrors the distinct roles of these two JNK isoforms in CD8 T cell activation. Studies by Conze and collaborators have shown that genetic abrogation of JNK1 impairs Ag-stimulated expansion of CD8 T cells in vitro and IL-2Rα (CD25) expression, whereas ablation of JNK2 positively regulates CD8 T proliferation and IL-2 production [95]. Furthermore, downregulation of CD25 expression in JNK1^−/−^ CD8^+^ T cells was associated with reduced levels of c-Jun and AP-1-binding activity [95]. These observations suggest that the differential regulation of c-Jun protein stability by JNK1 and JNK2 plays an instrumental role in the functional output of these two JNK isoforms in CD8 T cell activation.

### 2.6. Role of c-Jun N-Terminal Phosphorylation in c-Jun Biological Outputs

Although c-Jun phosphorylation is predominantly a proliferative and oncogenic event, it has also been postulated that c-Jun transactivation plays a contradictory role in apoptosis, either for or against, depending on the cell type and the corresponding environmental status of the cells [96]. c-Jun N-terminal transactivation induces apoptosis in NIH 3T3 fibroblasts and JNK-mediated multisite phosphorylation at Thr91/93/95 promotes apoptosis in cerebellar granule neurons [72,97]. On the other hand, phosphorylation of c-Jun at Ser63/73, along with the nuclear factor-kappa beta (NF-κB) transcription factor, protects from tumor necrosis factor (TNF)-α-mediated apoptosis [98]. Moreover, our studies in cerebellar granule cells, Bergmann glial cells and breast cancer cells have shown that the extent of c-Jun N-terminal multisite phosphorylation by JNK reflects the degree of extracellular signals, and plays an important role in determining the type of c-Jun biological output [72,73,99,100].

c-Jun and AP-1 protein complexes have lately emerged as components of the immune checkpoint regulation [49]. A list of co-stimulatory and co-inhibitory molecules of T-cell activation and T-cell efficient function are implicated as targets of promising clinical applications in several types of malignancy, such as metastatic melanoma, renal cell carcinoma, non-small-cell lung carcinoma (NSCLC) and Hodgkin’s lymphoma [49,101,102,103,104]. CD40 belongs to the group of co-stimulatory molecules and leads to the phosphorylation/activation of c-Jun, further augmenting production of interleukin-6 (IL-6) by B lymphocytes in a JNK/c-Jun-mediated manner [105]. On the other hand, in melanoma cells, phosphorylated/activated c-Jun, in synergy with signal transducer and activator of transcription 3 (STAT3) transcription factor, promotes the expression of the co-inhibitory molecule and anti-tumor immune checkpoint blocker PD-L1 by binding to the enhancer of its gene promoter [106]. Therefore, c-Jun, along with partner AP-1 proteins that have also been shown to participate in the transcriptional network that tends to restore T-cell function after immune checkpoint blockade, are potential therapeutic targets in anti-tumor immunotherapy [49]. 

### 2.7. Therapeutic Targeting of c-Jun/AP-1 Complexes

The crucial involvement of AP-1/c-Jun TFs in a plethora of biological processes and their abnormal activity in a wide range of human pathophysiologies, including dysregulated immune responses, indicate their potential as targets for therapeutic control. To this end, several pharmaceutical companies are currently focusing their efforts on creating low molecular weight compounds that specifically target c-Jun/AP-1 complexes, and will be easily given to patients with modest side effects. These small-molecule drugs may modulate the interaction of c-Jun/AP-1 complexes with cognate DNA sequences, coactivators/corepressors, or impede the dimerization, post-translational modifications (mainly phosphorylation) and degradation of AP-1 components, hence, altering their gene-regulatory effects.

Nevertheless, several challenges posed by c-Jun/AP-1 complexes as drug targets must first be surmounted. For instance, with regard to targeting c-Jun/AP-1–coactivator/corepressor protein–protein interactions, it is necessary to identify which of the cofactor complexes are the most clinically relevant to block. Furthermore, many of the individual components in AP-1 dimeric species exhibit substantial conformational plasticity i.e., exist in various ‘topological’ isomers. Although the latter increases the intricacy of c-Jun/AP-1 complexes as drug targets, it also provides a pharmacological prospect for augmenting drug selectivity and eliciting context-dependent effects.

The development of c-Jun/AP-1-specific small-molecule drugs for clinical application will also demand advanced diagnostic tools to identify those patients who harbor aberrant AP-1 TFs. Ongoing efforts to elucidate the molecular events underpinning the pathobiology of AP-1 TFs will offer clues on how fine-tuning of c-Jun/AP-1 functions by low molecular weight compounds can be achieved and, in turn, translated into therapeutic interventions in a spectrum of human disorders [107].

## 3. Role of c-Jun/Ap-1 in the Transcriptional and Epigenetic Regulation of Effector CD8T Cell Differentiation

### 3.1. The CD8 T Cell Response to Acute Infection

Antigen-specific naïve CD8 T cells respond to acute infection by undergoing extensive proliferation and differentiation to produce a copious number of cytotoxic T lymphocytes (CTL). The initial CTL pool is not homogenous and includes short-lived terminally differentiated effector cells (SLECs) and memory precursor effector cells (MPECs) [7,108,109,110]. SLECs are IL-2 dependent, highly cytotoxic and are rapidly recruited at the site of infection to produce high levels of cytotoxic molecules; then, they rapidly die after pathogen clearance. In contrast, MPECs develop in effector T memory cells and central memory T cells [111].

Antigen-specific CD8 T cell activation is dependent on two concomitant signals: the first initiated by the recognition of an antigen-MHC complex by the T-cell receptor (TCR) (signal 1); the second initiated by ligand-mediated activation of co-stimulatory receptors expressed on T cells (i.e., CD28) (signal 2) [112]. Although these two signals are sufficient to initiate the clonal expansion of CD8 T cells, full expansion of CTLs and differentiation in either SLECs or MPECs requires an additional signal (signal 3), which is initiated by autocrine Il-2 and environmental inflammatory cytokines, including IL-12 and type I interferons [109,113,114,115]. In particular, IL-12 inversely regulates the expression of T-bet and Eomes, two T-box containing TFs involved in differentiation of effector and memory CD8 T cells, respectively [109,116]. Furthermore, Joshi et al. have shown that the amount of IL-12 present during CD8 T cell priming regulates cell fate by inducing graded expression of T-bet; with high levels leading to maturation of SLECs and low levels allowing for the differentiation of MPECs [109]. The study by Agarwal et al. was the first to describe that the essential mechanism mediating the effect of IL-12 on CD8 T cell differentiation was chromatin remodeling, which allows continuous expression of genes transiently activated by TCR and costimulatory receptors [114]. Studies in individual CD8 T cells have confirmed that chromatin remodeling factors and lineage-specific transcription factor act together to direct a dynamic transcriptional network regulating the differentiation of CD8 T cell into effector and memory cells [19,117,118].

### 3.2. c-Jun/AP1 Cooperates with NFAT Downstream TCR and Costimulatory Signaling

NFAT proteins have been largely shown to be the major family of TFs involved in the regulation of antigen-dependent activation of T cells [5,119]. The NFAT family includes five members, of which four (NFAT1-4) are activated by intracellular Ca^++^ signaling. Briefly, NFAT activation is initiated by the TCR ligation, which leads to PLCγ-dependent production of DAG and InsP3, in turn resulting in the activation of the Ras/MAPK pathway and the cytosol release of ER—stored Ca^2+^, respectively. The release of ER-stored Ca^2+^ triggers the store-operated Ca^2+^ entry (SOCE), a pathway leading to the extracellular Ca^2+^ entry and the consequent activation of calcineurin [120]. Calcineurin dephosphorylates multiple substrates, among which the regulatory domain of cytosolic NFAT, thereby allowing NFAT nuclear translocation [121]. In turn, depending on concomitant signaling pathways, nuclear NFAT can cooperate with various TFs, leading to distinct T cell programs, as T cell activation, T cell anergy and T cell exhaustion [5]. As regard CD8 T cell activation, the AP-1 transcription factor, mainly c-Jun/c-Fos, is the major NFAT partner, with which it cooperates to induce the expression of effector genes [5].

Although the PLCγ/RAS/MAPK pathway induced by TCR ligation promotes the synthesis, phosphorylation and activation of AP-1 members, a costimulatory signal (signal-2) is required to further activate AP-1 [5,122,123]. The major costimulatory receptor present on CD8 T cell surface is CD28, a type I transmembrane glycoprotein expressed in both naïve and activated CD8 T cells [112]. Expression of CD28 ligands, the B7.1 (CD80) and B7.2 (CD86) surface adhesion molecules, is restricted to professional APC cells. The binding of CD80 and CD86 to CD28 triggers the association of intracellular adaptor proteins with the intracellular tail of CD28. Briefly, the membrane proximal YMNM motif and the distal PYAP motif interact with several kinases and adaptor proteins, mostly through their SH2 and/or SH3 domains [124,125]. These motifs are involved in the activation of NFAT, AP-1 and NF-kB TFs [124]. In particular, the adaptor protein GRB2 and the guanine nucleotide exchange factors (GEF) Sos and Vav1 interact with the PYAP motif via their SH2 and SH3 domains, respectively [125]. These interactions result in activation of small GTP-binding proteins Rac1 and CDC42, which, in turn, activate the JNK/c-Jun pathway [123,126,127].

Mechanistically, NFAT and c-Jun/AP-1 cooperate by forming ternary complexes on NFAT/AP-1 composite sites present in the regulatory regions of several effector genes (Table 2) (Figure 1B) [12,13,17,128,129,130,131]. Pioneer studies have shown that once bound, NFAT/AP-1 complexes induce the formation of nucleosome-free DNAase I hypersensitive sites (DHSs) at the promoter and enhancer, suggesting a role in the regulation of chromatin accessibility [128,132] Recent studies, using the high resolution ATAC-seq method to analyze global changes in chromatin accessibility [133], have then confirmed that NFAT/AP-1 complexes promote chromatin accessibility at TCR-inducible enhancers, hence enabling the recruitment of pre-existing TFs [14,123,134,135]. Furthermore, the genome-wide mapping of TF enriched motifs in accessible regions (DARs) of naïve versus activated CD8 T cells has confirmed that the NFAT/AP-1 composite sites are enriched at DARs of T cell effector genes [136].

Among NFAT/AP-1 target genes, the regulation of Il-2 expression has been extensively studied, as autocrine production of IL-2 is critical for different aspects of T cell biology, including clonal expansion [142,143]. Ishihara and Schwarz have proposed a two-step binding of TFs for the induction of high levels of IL-2 in T cells: a first step where the binding of NFAT-1/P-1 ternary complex promotes chromatin accessibility; a second step where the recruitment of acetyltransferase CREB-binding protein (CBP) by AP-1 allows histone modifications and the binding of later TFs, including RNApol II and TATAA binding protein (TBP) [144]. Interestingly, the pharmacological inhibition of JNK/c-Jun signaling impaired transcriptional activation (second step) but not chromatin accessibility (first step), suggesting distinct roles for phosphorylated c-Jun (pc-Jun) and unphosphorylated c-Jun (or JunB) in the regulation of Il-2 expression. Presumably, c-Jun/AP-1 or JunB/AP-1 cooperates with NFAT1 in promoting chromatin accessibility, whereas pc-Jun/AP1 recruits CBP on NFAT/AP-1 composite sites, thereby allowing the H3K27 acetylation and recruitment of TFs and RNA Pol II. Synergy between NFAT-1 and c-Jun/Ap-1 can also occur at a canonical NFAT site located in the proximal promoter of human IL-2 gene, through physical interaction with NFAT-1 [18]. Functionally, the synergy between c-Jun/AP-1 and NFAT is crucial for the maximal production of autocrine IL-2, which is required for a superior expansion potential of CD8T cells [137].

### 3.3. c-Jun Cooperates with BATF and IRF4 during Effector CD8 T Differentiation

Early after CD8 T cell activation, the acquisition of either SLEC or MPEC phenotypes is associated with chromatin priming, which consists in stable changes of chromatin accessibility and histone modifications in regulatory regions of genes associated with either phenotype [19,117,145]. Chromatin remodeling is initiated by upstream TCR-inducible “pioneer transcription factors” (PTF), which promote accessibility to regulatory regions containing binding motifs for constitutive TFs, as ETS and RUNX, or lineage-specific TFs [7,116,135,146,147]. Importantly, chromatin priming confers immunological memory by maintaining an epigenetic memory of previous gene activation [19,136].

IRF4, a member of the IRF family of transcription factors, is induced by TCR signaling, and plays a crucial role in lymphoid differentiation and immune response [148,149]. IRF4 binds weakly to DNA due to an intramolecular inhibitor domain; therefore, it requires cooperation with partner TFs to bind DNA at composite binding motifs present in the regulatory regions of target genes [150]. ChIP–Seq analyses in CD4+ T cells have shown that AP-1-IRF composite sites (AICE) direct the assembling of IRF4 with Jun/BATF heterodimers on a large array of genes required for differentiation of Th17 and Th2 cells, as well as B cells and DCs [151,152,153]. These findings brought to a view that cooperative binding of BATF/Jun and IRF4 at AICE motifs might function as a “pioneer transcription complex”, promoting chromatin accessibility at target genes [150,154]. Recent studies have confirmed that IRF4 cooperates with BATF and each of Jun proteins in inducing chromatin accessibility in regulatory regions of genes linked to effector differentiation of CD4+ T cells [19,116,135].

As regards CD8 T cells, Kurachi et al. have discovered that cooperation between IRF4 and BATF/Jun heterodimers occurs also in CD8 T cells (Figure 1B) [6]. IRF4, together with BATF and either of the Jun family members were shown to bind and promote the expression of genes expressed early during effector differentiation of CD8 T cells, as Tbx21 (coding for T-bet) and cytokine receptors [6]. In particular, expression patterns linked to CD8 T cell activation showed enrichment for genes bound by IRF4/BATF/Jun or IRF4/Jun. On the other hand, transiently expressed genes associated with cell-cycle progression were predominantly bound by c-Jun/AP-1 heterodimers, suggesting a critical for c-Jun in the clonal expansion of effector CD8 T cells [6]. In addition, ChIP-Seq analysis of the binding of BATF, IRF4 and each of the Jun proteins at the region with H3 histone modifications showed that all six TFs bind to active chromatin of gene loci involved in effector cell differentiation. These genes included Tbx 21 (coding for T-bet), Prdm1 (coding for Blmp-1) and IL-2Rα (coding for the IL-2 cytokine receptor CD25. Yet, c-Jun showed the highest frequency of TF peaks in active enhancers (marked by mono-methylation of H3K4 acetylation and acetylation of H3K27) and active promoters (marked by H3K4me3 and H3K27ac) [6]. This evidence suggests that c-Jun promotes chromatin remodeling at both enhancer and promoters.

Pioneer transcription factors (PTF) are believed to promote chromatin accessibility by recruiting at their binding motifs chromatin remodeling factors and histone modifier enzymes [134]. In this regard, Ito et al. have shown that Jun/Fos dimers bind with high affinity to the BAF60a subunit of the chromatin remodeling SWI-SFI complex and recruit it at AP-1 recognition motifs [155]. In addition, Ndlovu et al. have shown that c-Jun/Fra1 dimers promote chromatin accessibility at the IL-6 gene promoter by recruiting the Brg1 central ATPase subunit of SWI/SNF and the acetylases CBP/p300 at AP-1 binding motifs [156]. More recently, Vierbuchen et al. have shown that Jun/Fos dimers cooperate with cell-specific TFs by recruiting the SWI-SFI complex at enhancers [36]. These studies suggest that the cooperative binding of Jun/BATF with IRF4 at AICE composite sites facilitates the recruitment of chromatin-remodeling factors at regulatory regions of genes involved in CD8 T cell differentiation. In line with this hypothesis, recent studies, using ATAC-seq scanning of TF-binding motifs within accessible chromatin regions, have shown that binding sites for T-bet, BATF and AP-1 are depleted in naive CD8 T cells, but enriched in effector CD8 T cells [2,19,20].

## 4. Regulation of c-Jun/Ap-1 in Exhausted CD8T Cells during Chronic Viral Infection and Cancer

### 4.1. CD8 T Cell Exhaustion in Chronic Infection

During chronic infection, antigen-specific CD8 T cells initially acquire effector functions, which gradually disappear as the infection progresses. This loss of function is referred to as ‘exhaustion’ and is a feature common to most human chronic infections [157]. Multiple studies have shown that the CD8 T cell exhaustion state is a cell fate program initiated by the persistent exposure to pathogens and characterized by progressive loss of cytotoxic activity, diminished proliferative potential, altered preservation of memory and metabolic dysregulation [8]. The current understanding of CD8 T cell exhaustion considers this T cell response as a differentiation program activating preexisting feedback loops to limit cell damaging by cytotoxic effector molecules [158,159]. The first feature of this program is the continuous expression of multiple immune checkpoint inhibitor receptors (IRs), as PD-1, CTLA-4 and TIM-3, which are only transiently expressed during acute infection to prevent harmful overactivation [160]. In turn, the progressive increase of the CD8 T cell fraction expressing multiple IRs parallels a gradual decrease in the secretion of effector cytokines, including IL-2, TNFα and IFNγ [161,162]. In addition, exhausted CD8 T cells exhibit a specific expression profile characterized by the altered expression of several chemokines, as well molecules involved in chemotaxis, migration and metabolism [162].

Further studies have revealed that during chronic viral infection, the persistent exposure to viral antigens gives rise to a heterogeneous population of CD8 T cells, which comprises at least two major subsets of exhausted CD8 T cells: (i) one small subset of self-renewing progenitor exhausted CD8 T cells, marked by the expression of PD-1, transcription factor T-cell factor 1 (TCF1), T-bet, along with some effector cytokines; (ii) a larger subset of terminally differentiated exhausted CD8 T cells, derived from the progenitor pool and expressing high levels of multiple IRs, TFs involved in memory formation (Eomes and Blmp1), low levels of T-bet and TCF1, along with moderated levels of certain effector molecules (Figure 2) [163,164,165,166]. Although only progenitor exhausted CD8 T cell responded to anti-PD-1 therapy, both subsets were shown to cooperate in containing chronic viral infection, suggesting that the balance between renewal and terminal differentiation is crucial for the immune response to chronic infections [163]. Furthermore, Chen et al. found that a binary fate decision occurs early in chronic infection, which separates effector-like from exhausted CD8 T cell lineages, giving rise to: (i) a pool of T-bet^+^/PD-1^−^ short-lived cells with effector-like functions; (ii) a pool of TCF1^+^/PD-1^+^ cells with self-renewal property, which generates a pool of terminally differentiated exhausted cells [139]. This cell fate decision was largely regulated by TCF1, which suppressed T-bet expression and the development T-bet^+^ effector-like cells, while promoting TCF1^+^/PD-1^+^ exhausted progenitor cells (Figure 2). These finding are is in agreement with previous studies showing that TCF1 favors memory formation by repressing effector differentiation in CD8 T cells [140].

### 4.2. CD8 T Cell Exhaustion in Cancer

CD8 tumor infiltrating lymphocytes (TILs) were firstly shown to be highly cytotoxic lymphocytes exhibiting tumor-regression activity in human cancer patients [167,168,169,170,171]. Clinical studies have shown that the abundance of CD8 TILs is associated with good prognosis for several types of cancer [172,173,174,175]. In well-controlled tumors, the majority of TILs are effector CD8 T cells activated either in tumor-draining lymphnodes by antigen cross-presenting dentritic cells (DCs), or in the tumor by cross-reactive APC cells [176]. TILs are terminally activated and highly cytotoxic, expressing high levels of effector genes, as IFN-γ, Granzyme B and perforin [170]. On the other hand, TILs present in Hodgkin’s lymphoma and different types of human solid tumors express high levels of multiple IRs and are functionally impaired, suggesting that the persistent exposure of TILs to tumor-specific antigens leads to a exhaustion state similar to that observed during chronic viral infections [158,159,160,177,178,179,180].

Further studies have then demonstrated that the dysfunctional state of TILs is a dynamic program initiated early in tumorigenesis by persistent exposure to tumoral antigens and sharing phenotypic similarities and transcriptional pathways with exhausted virus-specific CD8 T cells [21,22,24,180,181,182,183]. In particular, Miller et al. have shown that the population of dysfunctional TILs from mouse melanoma is heterogeneous and acquires a state of exhaustion analogous to that induced by chronic viral infection [182]. Furthermore, they showed that TILs consist of a small subset of self-renewing TIF1^+^/PD-1^+^/TIM3^−^ progenitor exhausted cells, which exhibit polyfunctionality, long persistence and give rise to a large subset of short-lived TIF1^-^/PD1^+^/TIM3^+^ terminally exhausted TILs [182]. Notably, only the subset of progenitor exhausted TILs responded to anti-PD-1 therapy [182].

In addition to the studies discussed above and in the previous section, several other studies have contributed to a common view that persistent antigen exposure initiates a core epigenetic and transcriptional program in different settings of T-cell exhaustion (Figure 2), whereas genes and pathways unique for each setting reflect dissimilar, environment-specific, external signals [184]. As regards cancer, the complex tumor microenvironment (TME) contributes with a myriad of immunosuppressive factors to shape the antigen-driven core transcriptional profile in tumor specific CD8 T cells. Cancer cells and immunosuppressive cells, including tumor associated macrophages (TAMs), cancer associate fibroblasts (CAFs), myeloid derived suppressor cells (MDSCs), regulatory CD4T lymphocytes (Treg), may secrete a number of immunosuppressive cytokines in the TME. For example, TAM-derived TGFβ induces high expression of various IRs in TILs, which results in the repression of INF-γ and Granzyme B secretion in a dose-dependent manner [23]. Furthermore, the low availability of glucose, amino acids and oxygen in the TME, can suppresses the function of TILs by impeding metabolic changes that take place in T cells to meet the energetic demand [180].

### 4.3. The Core Transcriptional Program Driving Exhaustion is Triggered by Partnerless NFAT

Martinez et al. were the first to suggest that the early transcriptional pathway driving CD8 T cell exhaustion in chronic infection and cancer is initiated by NFAT activation in absence of AP-1 cooperation [21]. This study showed that a NFAT-1 variant unable to interact with c-Jun/AP-1 induces a transcriptional program establishing T cell exhaustion [21]. As discussed in Section 3, in effector CD8 T cells, NFAT and c-Jun/AP1 integrate TCR and costimulatory signaling by cooperating in the induction of effector cytokine (Figure 3A). In contrast, in virus-specific exhausted CD8 T cells or in exhausted TILs, the continuous expression of IRs spoils c-Jun/AP-1 activation, whereas the activation of NFAT remains unaffected. In turn, the lack of c-Jun/AP1-NFAT cooperation leads to a diminished expression of effector cytokines; whereas partnerless NFAT triggers a gene expression pattern associated to the exhaustion phenotype (Figure 3B,C) [21,162,185]. Genomic studies have shown that the pattern of chromatin accessibility specific for dysfunctional TILs is enriched for DNA binding motifs specific for NFAT and NR4A TFs; whereas transcription factor motifs for NFAT/AP-1 and canonical c-Jun/AP-1 are mostly associated with a pattern of chromatin accessibility specific of effector CD8 T cells [22].

Further studies have shown that, during chronic infection, NFAT, IRF4 and BATF are recruited to adjacent binding sites, and the binding of all three factors are significantly enriched among the core group of genes related to exhaustion, including Pdcd1 (encoding Blmp1), Lag3, Havcr2 (encoding TIM3) and CTLA4 [138]. These findings indicate that a NFAT/AP-1/IRF4 composite site (NAICE) directs the assembly of NFAT/BATF/IRF4 complexes at genes regulating CD8+ T cell exhaustion; whereas in effector CD8 T cells, the AICE composite site directs JUN/BATF/IRF4 complexes at genes regulating effector differentiation (Figure 3A,B) [6,138].

Moreover, Seo et al. have shown that NFAT induces the expression of secondary TFs, such as HMG-box transcription factor TOX and NR4A, which, in turn, increase the expression of multiple IRs and promote chromatin remodeling at genes driving exhaustion in CD8 T cells expressing chimeric antigen receptors (CAR T cells) CAR TILs [31]. Similarly, TOX was shown to act as a key inducer of epigenetic changes driving CD8 T cell exhaustion in chronic infection and cancer [25,26,27,28,29]. TOX was also shown to repress epigenetic events that are specific of effector cell differentiation (Figure 3B) [186]. These repressive events negatively regulated multiple TFs acting in networks downstream of TCR signaling, including AP-1 members c-Jun and Fos [25]. Furthermore, the levels of TOX proteins correlated with the severity of intra-tumoral CD8 T cell exhaustion in human cancers [186].

Taken together, these studies indicate that the core epigenetic and transcriptional program driving CD8 T cell exhaustion in chronic infection and cancer is initiated by the persistent Ag-dependent activation of TCR, which leads to two fundamental events (Figure 3B): (i) continuous induction of TCR-responsive IFR4, BATF and NFAT TFs; (ii) continuous expression of PD-1 and other IRs. In turn, the inhibition of costimulatory signaling by PD-1 (and other IRs) leads to a gradual loss of c-Jun/AP-1, so allowing partnerless NFAT to induce alternative target genes, including TOX and NR4A TFs. In sequence, TOX and NR4A promote changes in chromatin accessibility at enhancer and promoters of genes acting in the network of exhaustion; whereas TOX also represses epigenetic events at genes associated to effector differentiation. In parallel, the continuous increase of BATF and IRF4 expression, in the absence of c-Jun/AP-1, leads to a different partnership scenario, where NFAT/BATF/IRF4 form complexes at NAICE composite sites present in the enhancers and promoters of exhaustion-related genes. In addition, loss of c-Jun also favors the formation of JUNB/BATF/IRF4 complexes at the enhancer of exhaustion genes [30] (Table 2).

### 4.4. c-Jun as A Tool in CAR T Cell Therapy

Chimeric antigen receptor (CAR) T cell therapy is a major immunotherapeutic strategy aimed to reinvigorate patient own T cells. CAR T cells are genetically engineered to express chimeric antigen receptors that binds tumor antigens [187,188]. This therapy has resulted very effective against hematopoietic cancers, whereas it has not been very efficient against solid tumors, as CAR T cells become exhausted just like endogenous CD8 T cells responding to chronic infection or cancer [8,188]. To understand and possibly overcome the exhaustion program in CAR T cells, Lynn et al. established a mouse HA-28z CAR T cell model generating antigen-independent tonic signaling [30,189]. HA-28z CAR T cells exhibited a stronger exhaustion phenotype and showed significant changes in chromatin accessibility and gene expression patterns compared with control CD19-28z CAR T cells [30]. In particular, ATAC-seq data revealed that in HA-28z CAR T cells chromatin accessibility was linked to the enrichment of AP-1 and IRF binding motifs. In addition, the expression levels of JUNB, BATF3 and IRF4 were significantly increased compared to c-Jun, indicating that tonic signaling generated in HA-28z CAR T cells results in depletion of canonical c-Jun/c-Fos dimers. Next, the authors showed that HA-28z CAR T cells engineered to overexpress c-Jun restored the expression of Il-2 and IFNγ, increased functional capacity and enhanced anti-tumor activity in vivo. Additionally, c-Jun overexpression significantly altered the transcriptome of HA-28z CAR T cells, with an enrichment of AP-1 target genes [30].

Mechanistically, c-Jun DNA binding activity, but not its transcriptional activity, was necessary to overcome exhaustion, suggesting that c-Jun proteins displace JUNB/BATF3 and IRF4 transcriptional complexes at promoters or enhancer of exhaustion-associated genes [30]. Nevertheless, since c-Jun controls its own expression, this study corroborates the crucial role of costimulatory signaling in maintaining the adequate c-Jun levels to maintain chromatin accessibility at enhancers and promoters of effector genes. In line with this suggestion, previous studies have shown that the overexpression of NR4A1 displaces chromatin-bound c-Jun in dysfunctional T cells [141]. Conversely, CAR T cells deficient for all three members of the NR4A family (NR4A1, NR4A2 and NR4A3) were shown to suppress more effectively the growth of hCD19+tumors compared to wild-type (WT) CAR T cells [29]. Similarly, CAR T cells lacking the expression of both TOX and TOX2 promoted the regression of hCD19+ tumors more efficiently than WT CAR TILs or CAR TILs singly deficient in either TOX or TOX2 alone [31]. Furthermore, CAR T cells deficient in either TOX or NR4A reduced the expression of IRs and displayed greater chromatin accessibility at genomic regions enriched for motifs binding NF-κB and canonical AP-1 [29,31].

## 5. Conclusions

Physical interactions within a core set of signal-inducible TFs are essential for promoting chromatin accessibility at genomic loci driving CD8 T cell fate [7,118]. In particular, the ability of NFAT and IRF4 TFs to form ternary complexes with canonical Jun/AP-1 or Jun/BATF dimers, respectively, has emerged as a crucial feature enabling the integration of TCR and costimulatory signaling into patterns of gene expression driving CD8 T differentiation [6,29,116].

Moreover, the composition of these ternary complexes is dynamic, since context-specific signaling differentially regulates the expression levels of each Jun family member and BATF. In turn, the combinatorial interaction of Jun proteins and BATF with IRF4 results in distinct patterns of gene expression regulating CD8 T cell fate [6,30]. This scenario is in agreement with the current idea that cell type specification is actually accomplished by a network of TFs expressed uniquely in combination [116,190]. Recent insights into the exhaustion state of CD8 T cells suggest that persistent antigen exposure leads to an exhaustion gene-expression profile that is orchestrated by the unbalanced expression of c-Jun and TCR-inducible NFAT, BATF and IRF4 TFs (Figure 3B). In turn, the recruitment of NFAT/BATF/IRF4 complexes at exhausted-associated enhancer and promoters drives the expression of several genes characterizing the transcriptome profile of exhausted CD8 T cells [21,22,31,138]. Furthermore, loss of c-Jun expression in exhausted CAR TILs is accompanied by overexpression of JUNB, BATF3 and IRF4 TFs, in turn forming ternary complexes at gene loci involved in the exhaustion program [30]. Conversely, forced expression of c-Jun in exhausted CAR TILs restores the T cell effector phenotype and enhances anti-tumor activity in vivo [30]. Thus, in CD8 T cells, c-Jun/AP-1 seems to act as a guardian of anti-tumor activity: by forming ternary complexes with NFAT it prevents monomeric NFAT from initiating the transcriptional program of exhaustion; by displacing JunB and BATF3 from regulatory regions it blocks the expression of exhaustion-associated genes. However, the detailed mechanisms by which c-Jun counteracts CAR T cell exhaustion have yet to be elucidated, as are the long-term effects of c-Jun overexpression on the inflammatory transcriptome of CD8T cells. Nevertheless, the discovery of c-Jun as a possible promoter of exhaustion resistance offers innovative possibilities for improving immunotherapeutic strategies using the adoptive transfer of CAR T cells.

## Figures and Tables

**Figure 1 cells-09-02470-f001:**
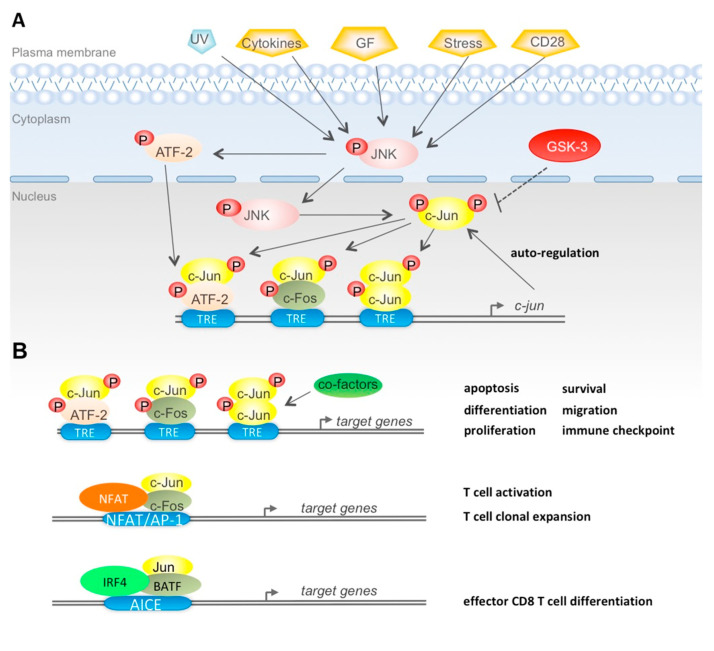
(**A**) c-Jun/activator protein-1 (AP-1) regulation and biological activity. (**A**) c-Jun/AP-1 regulation via phosphorylation: application of various extracellular stimuli (UV irradiation, cytokines, growth factors, stress and CD8 signaling) activates JNK of the MAPK pathway through phosphorylation. Activated JNK (p-JNK) potentiates c-Jun via phosphorylation at sites in the N-terminal domain, which triggers either homodimerization of c-Jun or heterodimerization with c-Fos. P-JNK also phosphorylates/activates ATF-2 which forms dimers with c-Jun. The dimers bind to TRE elements along with co-factors (not shown) in order to activate transcription of the *c-jun* gene, thereby setting up an auto-regulatory mechanism of c-Jun/AP-1. On the other hand, GSK-3 phosphorylates c-Jun at sites in the C-terminal domain, thus, reducing respective gene transcription. (**B**) c-Jun/AP-1 biological outputs: in different types of mammal cells, c-Jun/AP-1 binds to TRE elements, along with co-factors (co-activators or co-repressors), in order to activate transcription of genes that regulate proliferation, differentiation, apoptosis, survival, migration of normal and malignant cells, as well as immune checkpoint function for cells of the immune system (upper scheme). In T cells, c-Jun/AP-1 forms ternary complexes with NFAT and binds NFAT/AP-1 composite sites present in the regulatory regions of cytokines and effector genes (middle scheme). In CD8 T cells, Jun/BATF form ternary complexes with IRF4 and AICE composite sites present in the regulatory regions of genes controlling effector differentiation of Ag-activated CD8 T cells (lower scheme).

**Figure 2 cells-09-02470-f002:**
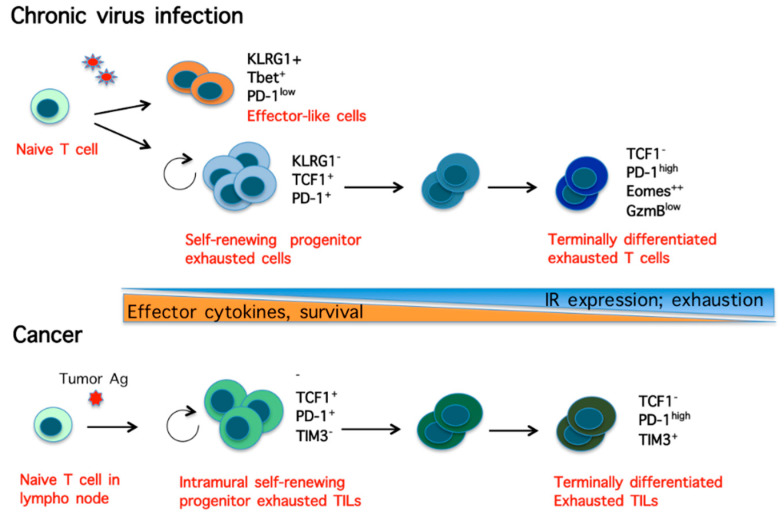
Heterogeneity of CD8 T cells during chronic viral infection and cancer. (**Upper scheme**) In chronic infection, virus-specific exhausted CD8 T cells consists of a population of T-bet^+^/PD-1^−^ short-lived effector-like T cells; a population of self-renewal transcription factor (TCF)1^+^/ programmed cell death protein 1 (PD-1)^+^ progenitor exhausted T cells, which generates a poll of cells progressively differentiating in terminally differentiated exhausted TCF1^−^/PD-1^high^/ T-cell Immunoglobulin domain and mucin domain 3 (TIM3)^+^ cells. (**Lower scheme**) Intramural tumor infiltrating lymphocytes (TILs) consist of a self-renewing TIF1^+^/PD-1^+^/TIM3^−^ cell population that gives rise to a pool of cells progressively differentiating in terminally differentiated TIF1^−^/PD1^+^/TIM3 cells.

**Figure 3 cells-09-02470-f003:**
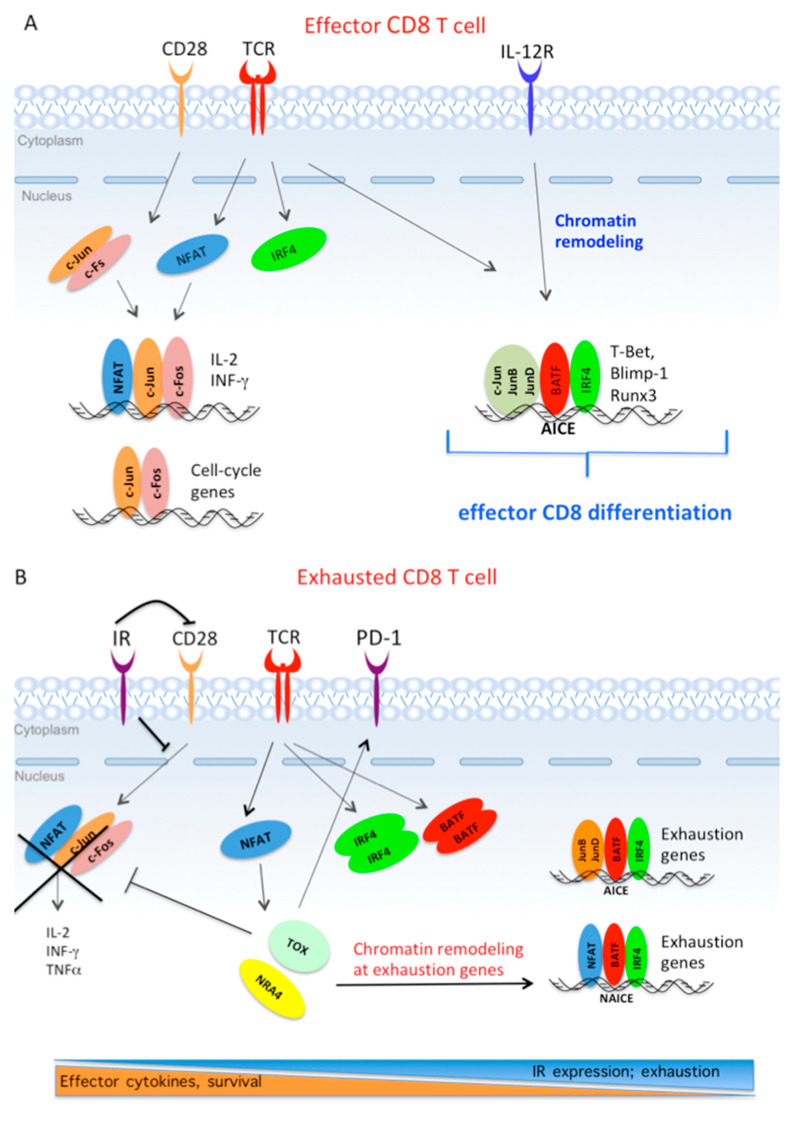
The early antigen-driven transcriptional pathway in effector CDT8 T cells and in exhausted CD8 T cells. (**A**). During acute infection, nuclear factor of activated T cells (NFAT)/c-Jun/c-Fos ternary complexes act downstream of T-cell receptor (TCR) and CD28 costimulatory receptor to induce the expression of effector cytokines. c-Jun/c-Fos also induce the expression of cell cycle genes. TCR-responsive ATF-like (BATF) and interferon responsive factor 4 (IRF4) TFs form ternary complexes with each of the Jun proteins at AP-1-IRF composite sites (AICE) composite sites present in enhancers of target genes, where they promote chromatin accessibility. IL-12R activation triggers chromatin remodeling, which allows continuous expression of lineage-specific transcription factors and effector genes. (**B**) During chronic infection and cancer, persistent antigen exposure leads to continuous expression of PD-1 and other inhibitor receptors (IRs), and to continuous induction of IFR4, BATF and NFAT TFs. The gradual loss of c-Jun/AP-1 by co-inhibitory signaling allows partnerless NFAT to induce TOX and NR4A TFs. In sequence, TOX and NR4A promote chromatin accessibility at exhaustion-associated genes. In absence of c-Jun, NFAT/BATF/IRF4 and JUNB/BATF/IRF4 form ternary complexes at NAICE and AICE composite sites present in enhancer and promoters of exhaustion-related genes.

**Table 1 cells-09-02470-t001:** c-Jun post-translational modifications.

PTM	Modified Site	Modifying Enzyme	Functions	REF
Phosphorylation	Ser63; Ser73	JNK1-2; ERK1-2	Transcritpional activation	[68,69,70]
Phosphorylation	Thr91; Thr93	JNK1-2-3	Transcriptional activation	[70,71]
Phopshorylation	Thr95	ND	Primimg Th93 phosphorylation	[72,73]
Phosphorylation	Thr239	GSK3	Inhibition of DNA binding	[70,74]
Phosphorylation	Thr243	ND	Priming 239Thr phoshorylation	[70,75]
Phosphorylation	Thr249	Casein kinase II	Inhibition of DNA binding	[76]
Dephopshorylation	Thr243	Calcineurin	Increase of protein stability	[77]
Ubiquitinotion	ND	Fbw7	Ubiquitin-dependent degradation	[78,79]
Acetylation	Lys271	p300/CBP	Transcriptional repression	[64]
Sumoylation	Lys229	Pias1; Piasxb	Transcriptional repression	[65,66]

**Table 2 cells-09-02470-t002:** Transcriptional regulatory network of NFAT/AP-1/IRF4.

Gene	Function in T cells	Transcription Factors	CD8 T Cell Differentiation State	RREF
IL-2	Clonal expansion	NFAT/AP-1(c-Jun/c-Fos) complex	Effector	[4,17,18,126,137]
INFg	Effector cytokine	NFAT/AP-1(c-Jun/c-Fos) complex	Effector	[12,13]
TNFa	Effector cytokine	NFAT/AP-1(c-Jun/ATF2) complex	Effector	[130]
Granzyme B	Cytotoxic	NFAT-2	Effector/CTL	[131]
Pdcd1 (PD-1)	Immunoregulatory receptor	NFAT; BATF; NFAT/BATF/IRF4 complex	Exhaustion	[21,138]
Havcr2 (TM3)	Immunoregulatory receptor	NFAT/BATF/IRF4 comlex	Exhaustion	[138]
Tbx21 (T-bet)	Lineage-specific TF	Jun/BATF/IRF4 complex	Effector	[6]
Tcf7 (TCF1)	Lineage-specificTF; self-renewal	Repressed by Blmp1	Memory; Progenitor- exhausted cells	[139,140]
Prdm1 (Blmp1)	Transcritional repressor	Jun/BATF/IRF4 complex	Memory; Exhaustion	[6]
Eomes	Lineage-specific TF	Jun/BATF/IRF4 complex	Memory; Exhaustion	[6]
TOX	TF; chromatin accessibility	NFAT	Exhaustion	[25,31]
NR4A	TF; chromatin accessibility	NFAT	Exhaustion	[29,141]

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
