# Peer review of "The Multifaceted Output of c-Jun Biological Activity: Focus at the Junction of CD8 T Cell Activation and Exhaustion"

_cells, 2020, doi:10.3390/cells9112470_

Round 1

Reviewer 1 Report

The review titled "The multifaceted output of c-Jun biological activity: focus at the junction of CD8 T cell activation and exhaustion" focuses on the multi-functionality of the c-jun, a member of the Activator Protein-1 (AP-1) transcriptional complex. The review is focused and is very well written. Includes recent studies and covers most of the known data on the signalling pathways involved.  They describe briefly the various levels of regulation of c-jun under different cellular processes.  The review focuses mainly on the regulatory role of c-jun on their differentiation and exhaustion processes of CD8 T cells during chronic infection and cancer.  This information could help in developing targeted immuno therapies for cancer. This review has the potential to attract researchers working on signaling pathways.  

However, the review needs to modified substantially before it could be accepted for publication.

The major concerns are

1) Abbreviations should not used in the abstract. Please modify it accordingly

2) The introduction needs to rewritten. It looks more like a discussion than giving an overview of what the review is about and its significance. Make it short and concise. Write in simple sentences so that it can easily connect with the readers. 

Please stick to writing the full form rather than using abbreviations when using it first time in the document. Modify wherever is required.

3) Please include more figures illustrating the role of c-jun in CD8 T cells as well in chronic infection and cancer

4) Please try putting sub-titles under major titles which will make the review more detail and organised. All sections are written very elaborately and seem to lose focus towards the end of each section which needs to be taken care of.

5) Try including 1 or 2 tables and it will also help to organize better.  

Minor concerns

1) Typographical errors and grammatical errors need to corrected throughout the text.

Author Response

Reviewer 1:

We appreciated very much the constructive criticism of the reviewer. All comments and suggestions were very helpful and greatly improved the quality of our manuscript.

Modified or new text is highlighted in green.

Comment 1: Abbreviations should not used in the abstract. Please modify it accordingly.

Response: We have substituted AP-1, bZIP ,ATF, BATF and JDP2 with full forms.

Comment 2: The introduction needs to rewritten. It looks more like a discussion than giving an overview of what the review is about and its significance. Make it short and concise. Write in simple sentences so that it can easily connect with the readers. 

Please stick to writing the full form rather than using abbreviations when using it first time in the document. Modify wherever is required.

Response : We have appreciated these useful criticisms, accordingly we have rewritten the introduction in a more concise manner.

Also, we have used full form accordingly to your request.

Comment 3: Please include more figures illustrating the role of c-jun in CD8 T cells as well in chronic infection and cancer.

Response: We fully agree with this comment and have included two more figures:

Figure2 illustrating the heterogeneity of exhausted CD8T cells during chronic viral infection and cancer

 Figure3-A illustrating the role of c-Jun in the antigen-driven transcriptional pathway of effector CD8 T cells; figure 3B illustrating the antigen-driven transcriptional pathway occurring (in absence of c-Jun) in exhausted CD8 T cells.

Comment 4: Please try putting sub-titles under major titles which will make the review more detail and organised. All sections are written very elaborately and seem to lose focus towards the end of each section which needs to be taken care of

Response: We fully agree with this comment and thank the reviewer for having pointed it out. We have reorganized all main sections in sub-paragraphs, which helped to keep their focal points in place. We also wrote new text in section 4 to discuss in more details the heterogeneity of exhausted CD8T cells (section 4.2), and the core transcriptional pathway driving CD8 T cell exhaustion both in chronic infection and cancer (section 4.3).

Comment 5: Try including 1 or 2 tables and it will also help to organize better

Response: We fully agree with this comment and have made two tables: Table1 listing the main post-translational modifications of c-Jun; Table 2 listing examples of NFAT, NFAT/AP-1 and JUN/BATF/IRF4 target genes that are associated with distinct differentiation states of CD8 T cells.

Minor comments: Typographical errors and grammatical errors need to corrected throughout the text.

Response: we have made corrections throughout the text

Reviewer 2 Report

This is a well written comprehensive review on the role of c-jun transcription factor on T cell activation and differentiation. The authors have nicely summarized various studies and described how c-jun can integrate a variety of signals by forming heterodimers with different transcription factors to achieve various signals by forming ternary complexes with different transcription factors in gene expression of T cells.

However, I request that the authors do the following, for purposes of easing quicker-data assimilation by the reader.

  1. The TFs, of AP1 family, Jun , fos etc, are discussed in terms of phosphorylation many times as they must, a table indication kinase, phosphatase, target site (# of the phosphor-acceptor T/S/Y) indicated, affect increase decrease activity, for all members.
  2. The same type of table but for any other modification, i.e. Ub, sumoylation, acetylation, etc.
  3. The same type of table but for all known miRNA affecting Jun-fos-MAF etc.
  4. A table listing the target genes that control T cell functions discussed in the review.
  5. Some small discussion on the possibilities, if any, of therapeutic control, of these TFs.

We thank the authors for an excellent job.

Author Response

Reviewer 2

We appreciated very much the constructive criticism of the reviewer. Comments were very helpful in improving the quality of the manuscript, which we now re-submit as a revised version including new next revisions, new figures and tables. We hope that you will now consider our study suitable for publication.  Modified or new text is highlighted in green

Comment 1-2: The TFs, of AP1 family, Jun , fos etc, are discussed in terms of phosphorylation many times as they must, a table indication kinase, phosphatase, target site (# of the phosphor-acceptor T/S/Y) indicated, affect increase decrease activity, for all members.

The same type of table but for any other modification, i.e. Ub, sumoylation, acetylation, etc.

Response: We fully agree with this comment.  However, since the review is focus primarily on c-Jun, we limited the Table (Table1) to c-Jun, listing all post-translational modifications following your instructions. 

Comment 3: The same type of table but for all known miRNA affecting Jun-fos-MAF etc.

Response: We believe that in section 2 of the review we have discussed in details the most significant aspects of c-Jun/Ap-1 regulation, including the regulation of c-Jun expression by miRNA and lncRNA. Considering that the main focus of the review is the role of c-Jun in the immune response of CD8T cells, we focused more on gaining insights into the studies defining the epigenetic and transcriptional programs that regulate CD8 T cell differentiation.

Comment 4: A table listing the target genes that control T cell functions discussed in the review.

Response: We have appreciated these useful criticisms. Accordingly, we made Table 2 listing examples of NFAT, NFAT/AP-1 and JUN/BATF/IRF4 target genes that are associated with distinct differentiation states of CD8 T cells.In addition, we made a figure (Figure3) illustrating the role of c-Jun in the antigen-driven transcriptional pathway of effector CD8 T cells, as well as the antigen-driven transcriptional pathway occurring in in exhausted CD8 T cells (in absence of c-Jun).

Comment 5: Some small discussion on the possibilities, if any, of therapeutic control, of these TFs.

Response: We thank the reviewer for this useful criticism, accordingly we have written a new paragraph (2.7).

Reviewer 3 Report

In this review the authors focus on the transcriptional and epigenetic regulatory mechanisms where the c-Jun/AP-1 transcription factors modulate the productive immune response of CD8 T cells, and how its deregulation in tumor reactive CD8 T cells, disrupt T cell function. The authors also highlight JNK/c-Jun pathway as possible target for immunotherapy-based combination aimed to stimulate anti-tumor immune functions.

The manuscript describes deeply the functions and regulation of c-Jun/AP-1 transcription factors, and it is shown an accurate scheme in figure 1, that makes easier to understand the complexity of the mechanisms activated by this pathway. Next the authors target how c-Jun/Ap-1 are involved in the regulation of CD8T cell differentiation and they ends characterizing  the role of these transcription factors in exhausted CD8T cells mainly in chronic infection and cancer.

The review provides in detail the function and regulation of c-Jun on CD8T cell activation and exhaustion. The article is clear and well structured.

Minor reviews:

Paragraph :

2.3. Regulation of c-Jun biological activity by N-terminal and C-terminal phosphorylation

  • In line 269 is written that “JNK2 is the main isoform to target c-Jun for Fbw7-dependent polyubiquitination in the brain (80). “ But in the reference 80 is not described in any part that JNK2 specifically is responsible for it or that it has this role. I’d be necessary to provide more o another reference to assert this information.

3.2. c-Jun cooperates with BATF and IRF4 during effector CD8 T differentiation

  • In line 466 it is not written the meaning of PTFs, and I couldn’t find in the manuscript. I’d be better if the authors can specify the PTFs acronyms
  • In line 472 there is a grammatical mistake: “ have shown etthat Jun/Fos dimers” I understand that the correct sentence has to be “have shown that Jun/Fos dimers”.
  1. Regulation of c-Jun/Ap-1 in exhausted CD8T cells during chronic infection and cancer
  • In line 495 there is a misspelling : “by progressives loss od cytotoxic activity” . I suggest to change for of.
  • In line 583 is written twice “either”, one has to be deleted.

Author Response

Reviewer 3

We appreciated very much the constructive and useful criticism of the reviewer. We now re-submit as a revised version including new next revisions, new figures and tables.Modified or new text is highlighted in green

Comment 1.  Regulation of c-Jun biological activity by N-terminal and C-terminal phosphorylation: In line 269 is written that “JNK2 is the main isoform to target c-Jun for Fbw7-dependent polyubiquitination in the brain (80). “ But in the reference 80 is not described in any part that JNK2 specifically is responsible for it or that it has this role. I’d be necessary to provide more o another reference to assert this information.

Response: We fully agree with this comment. In fact, in the study by Nateri was not specified. The sentence was confusing, so we rewrote the following sentence

«....... suggestig that JNK2 might be the main JNK isoform targeting nuclear c-Jun for Fbw7-dependent polyubiquitination inthe brain. However, more studies are required to elucidate which JNK isoform targets c-Jun for ubiquitin-dependent degradation in neurons.»

Comment 2  

- In line 466 it is not written the meaning of PTFs, and I couldn’t find in the manuscript. I’d be better if the authors can specify the PTFs acronyms.

Response: Thank you for this comment.  Actually it was specified in lane 439, presumably it’s better for the reader to specify one more.

  • In line 472 there is a grammatical mistake: “ have shown etthat Jun/Fos dimers” I understand that the correct sentence has to be “have shown that Jun/Fos dimers”. In line 495 there is a misspelling : “by progressives loss od cytotoxic activity” . I suggest to change for of.
  • In line 583 is written twice “either”, one has to be deleted.

Response: Thank you very much for pointing these mistakes out! We corrected them

Reviewer 4 Report

The authors review the current knowledge of how  c-Jun/AP-1 regulates distinct cellular responses specially focus  in the response of CD8 T cells to acute infection and cancer. They discuss the epigenetic regulatory mechanisms by which c-Jun/AP-1 may act to induce a productive immune response of CD8 T cells, and how c-Jun/AP-1 down-regulation may lead to CD8 T cell exhaustion. This subject is of interest and I recommend its publication .

Author Response

We thank the reviewer very much for the positive evaluation of our manuscript

Round 2

Reviewer 1 Report

My comments have been very well addressed and I am happy with the edited version of the review. 

A few minor suggestions:

1) Please add in the keywords: CD8 T cell exhaustion, CAR T cell therapy, acute infection, cancer

2) Line 592, it is NFAT-1 and not NFAF-1.